# Calcium Carbonate Nanoparticles—Toxicity and Effect of In Ovo Inoculation on Chicken Embryo Development, Broiler Performance and Bone Status

**DOI:** 10.3390/ani11040932

**Published:** 2021-03-25

**Authors:** Arkadiusz Matuszewski, Monika Łukasiewicz, Jan Niemiec, Maciej Kamaszewski, Sławomir Jaworski, Małgorzata Domino, Tomasz Jasiński, André Chwalibog, Ewa Sawosz

**Affiliations:** 1Department of Animal Breeding, Institute of Animal Sciences, Warsaw University of Life Sciences (WULS–SGGW), 02-787 Warsaw, Poland; monika_lukasiewicz@sggw.edu.pl (M.Ł.); jan_niemiec@sggw.edu.pl (J.N.); 2Department of Ichthyology and Biotechnology in Aquaculture, Institute of Animal Sciences, Warsaw University of Life Sciences (WULS–SGGW), 02-787 Warsaw, Poland; maciej_kamaszewski@sggw.edu.pl; 3Department of Nanobiotechnology, Institute of Biology, Warsaw University of Life Sciences (WULS-SGGW), 02-787 Warsaw, Poland; slawomir_jaworski@sggw.edu.pl (S.J.); ewa_sawosz@sggw.edu.pl (E.S.); 4Department of Large Animal Diseases and Clinic, Institute of Veterinary Medicine, Warsaw University of Life Sciences (WULS–SGGW), 02-787 Warsaw, Poland; malgorzata_domino@sggw.edu.pl (M.D.); tomasz_jasinski@sggw.edu.pl (T.J.); 5Department of Veterinary and Animal Sciences, University of Copenhagen, Groennegaardsvej 3, 1870 Frederiksberg, Denmark; ach@sund.ku.dk

**Keywords:** calcium carbonate, nanoparticles, chicken embryo, broiler, bone quality

## Abstract

**Simple Summary:**

Intensive selection in broiler chicken flocks has led do several leg disorders. The injection of nanoparticles, with high specificity to the bone, into the egg is a potential method to improve bone quality. The objective of our study was to evaluate the potential effect of calcium carbonate nanoparticles injected to the egg on chicken embryo development and bone quality of broiler chickens after 42 day of life. The calcium carbonate nanoparticles were not toxic to embryo and even improved the bone quality of embryos and later broilers without negative impact on production results. Thus, the application of calcium carbonate nanoparticles to the egg may be the potential solution for improving the bone mineralization of broiler chickens.

**Abstract:**

The use of intensive selection procedure in modern broiler chicken lines has led to the development of several skeletal disorders in broiler chickens. Therefore, current research is focused on methods to improve the bone quality in birds. In ovo technology, using nanoparticles with a high specificity to bones, is a potential approach. The present study aimed to evaluate the effect of in ovo inoculation (IOI) of calcium carbonate nanoparticles (CCN) on chicken embryo development, health status, bone characteristics, and on broiler production results and bone quality. After assessing in vitro cell viability, the IOI procedure was performed with an injection of 500 μg/mL CCN. The control group was not inoculated with CCN. Hatchability, weight, and selected bone and serum parameters were measured in embryos. Part of hatchlings were reared under standard conditions until 42 days, and production results, meat quality, and bone quality of broilers were determined. CCN did not show cytotoxicity to cells and chicken embryo and positively influenced bone parameters of the embryos and of broilers later (calcification) without negatively affecting the production results. Thus, the IOI of CCN could modify the molecular responses at the stage of embryogenesis, resulting in better mineralization, and could provide a sustained effect, thereby improving bone quality in adult birds.

## 1. Introduction

Currently, broiler lines are intensively selected for higher growth rate and increased final body weight (BW) [1]. However, this rapid weight gain in fast-growing chickens leads to several pathologies in the bones, such as osteoporosis and deformations in the legs [2]. This results in high economic losses, because skeletal abnormalities are one of the most common problems in modern chicken industries and cause reduction in consumed meat quality [3]. Moreover, broken legs lead to decreased feed intake, which eventually results in reduced final BW [4]. Bone strength in broiler legs is strictly associated with mineralization, especially Ca deposition, which is 99% localized in the bones [5,6]. This macroelement contributes to the mineral structure of the bone with P to form hydroxyapatite. In addition, Ca widely contributes to many essential physiological processes such as cell proliferation, muscle contraction, blood coagulation, enzyme activation, and hormone secretion [6,7]. Among all the Ca sources, limestone, a naturally occurring mineral primarily composed of calcium carbonate (CaCO_3_), is the predominant Ca supplement used in broiler production [8]. The inorganic forms of elements, such as oxides, sulfates, and carbonates, are widely used in poultry nutrition, mainly for economic reasons [9]. The concentration and bioavailability of Ca can differ according to the source or particle size [8,10]. Currently, there is a growing interest in the application of nanotechnology to produce a supplemental source of minerals in poultry diets. The use of nanoparticles as sources of various elements in poultry nutrition is a response to today’s consumption trends and society expectations. Minerals administered in the nanometric size can improve the welfare and health of animals because of their better availability [11]. Moreover, the aspect of environmental protection is also important—nanoparticles, as an innovative alternative to conventional sources of minerals, can be better absorbed by animals, thus reducing the excretion of minerals [12,13,14]. Calcium compounds in nanoparticle forms have a high prospective in poultry production [15]; so far, however, research is limited. Most studies are primarily concerned with per os administration of Ca nanoparticles [13,16,17,18]. The supplement of calcium compounds in the form of nanoparticles is also considered as a strategy to bring down the cost of calcium (and phosphorus) supplement in the feed [16]. The IOI of nutrients remains an interesting, alternative method of providing functional nutrition using various compounds. Nutrient supplementation by IOI could be more efficient when a selected compound is attached to nanoparticles, because of their ability to effectively deliver the compound inside cells and body tissues due to their small size (1–100 nm) [19]. According to Sawosz et al. and Zielinska et al. [20,21], the administration of nanoparticles directly to the egg at the early stage of embryogenesis can lead to a number of molecular and systemic changes. This, in turn, can enable a “better start” for newly hatched chickens and then influence the health and production status of the birds at later stages of life. For example, Mroczek-Sosnowska et al. [22] suggested that Cu nanoparticles administered in ovo interfere with the molecular status of muscle maturation during embryogenesis via MyoD1 and Pax7 proteins and later actually proved that breast muscle was bigger in broilers [23]. In line with this approach, we hypothesized that calcium carbonate (CaCO_3_) nanoparticles (CCN) as a safe and nontoxic supplement can be delivered by IOI at the early stage of incubation. They can exert several stimulative and modulatory effects on the development of the chicken embryo and then influence the bone quality of broiler without affecting production results. CCN as an external source of Ca may affect the regulation of bone osteocalcin (OC), the protein responsible for hydroxyapatite binding, ultimately resulting in increasing bone mineralization. Studies focusing on IOI of calcium compound nanoparticles are very limited [24,25,26]. The present study aimed to evaluate the effect of the IOI of CCN on chicken embryo development, health status, bone characteristics, and production results as well as meat quality and bone quality of broiler chickens after 42 days of rearing.

## 2. Materials and Methods

### 2.1. Nanoparticle Characterization and Preparation

CCN (white nanopowder, 97.5%) were obtained from SkySpring Nanomaterials, Inc. (Houston, TX, USA). Hydrocolloids of CCN of increasing concentrations (5, 10, 25, 50, 100, and 500 μg/mL) were produced by mixing the nanopowder with ultrapure water. After premixing the whole solution, ultrasound was introduced into the solution for 45 min by using Ultron U-509 apparatus (Transfer Multisort Elektronik, Lodz, Poland). The average zeta potential and particle size determination was carried out using Zetasizer Nano-ZS ZEN 3600 (Malvern Instruments Ltd., Malvern, UK) with the dynamic light scattering mode and laser Doppler electrophoresis at room temperature (23 °C). The size and shape of a single CCN were visualized using a Morgagni 268D transmission electron microscope with a wide-angle Olympus Morada digital camera (FEI, Hillsboro, OR, USA). The individual CCN hydrocolloids were prepared 30 min before the specific procedure was performed in each part of the experiment.

### 2.2. In Vitro Cytotoxicity and Mineralization

#### 2.2.1. Cell Isolation

Fertilized chicken eggs (*Gallus gallus domesticus*; *n* = 20) were supplied by a commercial, certified hatchery. On the twelfth day of incubation, the embryos were sacrificed, and their femurs were collected to obtain the primary osteogenic cells using modified method for bone cells isolation described by Li et al. [27]. The bones were cleaned of soft tissues; the bone shaft was isolated, crushed mechanically, and treated with collagenase for 30 min. The suspension was filtered through a 74-μm mesh sieve and centrifuged at 200× *g* for 10 min at room temperature. The supernatant was discarded, and the pellet was resuspended in medium containing DMEM and 10% (v/v) PBS.

#### 2.2.2. Viability Assay

Cell viability was evaluated using a 2,3-bis-(2-methoxy-4-nitro-5-sulfophenyl)-2H-tetrazolium-5-carboxyanilide salt (XTT)-based cell proliferation assay kit (Life Technologies, Taastrup, Denmark). Cells were plated in 96-well plates (5 × 10^3^ cells per well) and incubated for 24 h. The medium was then removed, and CNN samples at concentrations of 5, 10, 25, 50, and 100 μg/mL were introduced into the medium. Next, 50 μL of XTT solution was added to each well and incubated for an additional 3 h at 37 °C. The optical density of each well was recorded at 450 nm by using an enzyme-linked immunosorbent assay reader (Infinite M200, Tecan, Durham, NC, USA). Cell viability was expressed as a percentage (OD_test_ − OD_blank_)/(OD_control_ − OD_blank_) × 100, where OD_test_ is the optical density of cells exposed to CCN, OD_control_ is the optical density of the control sample, and OD_blank_ is the optical density of wells without cells.

#### 2.2.3. Cell Staining

Primary osteogenic cells were seeded in six-well plates (1 × 10^5^ cells per well) and incubated for 24 h. Cells cultured in the medium without the addition of CCN were used as the control. CCN were added to the cells at increasing concentrations (5, 10, 20, 50, and 100 mL/L). After 24 h, the cells were fixed with 4% paraformaldehyde and stained with a 2% Alizarin red solution (Merck, Warsaw, Poland) [28]. Cell morphology was recorded using an optical microscope (TL-LED, Leica Microsystems, Wetzlar, Germany).

### 2.3. In Ovo Experimental Design

According to 3rd Local Ethics Committee for Animal Experiments in Warsaw University of Life Sciences, the experiments on chicken embryo and broilers followed the approval of Local Ethics Committee (Approval No. 46/2015). The experimental material consisted of 240 fertilized chicken eggs from 37-week-old Ross × Ross 308 hens. First, the eggs were stored in a refrigerator at 12 °C and 73% humidity for 2 days and then placed in an incubator (Jamesway, Cambridge, ON, Canada). The eggs were weighed (55.9 ± 1.83 g) on day 1 of incubation and randomly divided into two groups, with 120 eggs per group. The control group was not inoculated, and the experimental group was supplemented with 500 μg/mL hydrocolloid of CCN in 0.2 mL volume per egg. Negative control (inoculated with PBS) was not included in this study. Before the injection procedure, the eggs were immersed in a 0.5% solution of potassium permanganate. The hydrocolloid was inoculated on the first day of incubation under sterile conditions into the albumen, using 27-gauge, 20-mm needles. The hole was sealed using a sterile tape, and the eggs were placed in an incubator immediately after the injection. Standard incubation conditions were provided to all eggs (temperature 37.8 °C, humidity 55%, turned once per hour for the first 18 days; 37 °C and 75% humidity on days 19 and 20). During incubation, the eggs were candled on the 7th and 18th d to discard unfertilized eggs and determine the dead embryo percentage. The hatchability was calculated as the ratio of eggs with living embryos on day 20 to the number of fertilized eggs in each group.

On day 20, eggs from each group were randomly selected, and all embryos were weighted and decapitated. During decapitation, pooled blood from 1.5 embryos (about 1.5 mL per sample) was collected (*n* = 10) from the jugular artery into glass tubes, stored at 4 °C overnight, and then centrifuged for 5 min at 1200× *g* (MPW-350R centrifuge, MPW Med. Instruments, Poland). The obtained serum was stored in cryovials at −80 °C. Previously selected embryos (10 per group) were immediately transferred on dry ice after blood collection. After cooling, selected organs were collected (liver, muscle, and both femurs and tibias), weighed, and stored at −80 °C. IOI procedure and the minimum necessary sample size estimation were performed with minor modifications as described by Lukasiewicz et al. and Pineda et al. [29,30].

### 2.4. Embryo Serum Biochemical and Toxicity Analyses

Ten serum samples from each group were analyzed by standard laboratory procedures in the Veterinary Diagnostic Laboratory at Warsaw University of Life Sciences by using commercial kits. Selected biochemical kits were used to detect the level of liver enzymes, including aspartate aminotransferase (AST), alanine aminotransferase (ALT), alkaline phosphatase (ALP), and lactate dehydrogenase (LDH), as well as kidney-related biochemical factors such as calcium, phosphorus, and creatinine. The levels of glucose (Glu), total protein (TP), albumin (Alb), total cholesterol (TC), and triglyceride (TG) were determined. Glutathione (GSH) was measured quantitively using Ellman’s method modified by Matusiewicz et al. [31]; in this method, 5,5ʹ-dithiobis(2-nitrobenzoic acid) (DTNB, Ellman’s reagent) is reduced by thiol compounds to form a colored product (2-nitro-5-mercaptobenzoic acid) with the maximum absorbance at 412 nm. To 375 µL of serum samples from each group, 19.7 µL of 50% trichloroacetic acid (TCA) was added and centrifuged (1200× *g*, 5 min). Then, 6.25 µl of deproteinized supernatants was transferred into a microplate and mixed with 50 µL of 0.2 M phosphate buffer (PBS) and 6.25 µL of 6 × 10^−3^ M DTNB. The absorbance was measured using Tecan’s NanoQuant Infinite M200 PRO (Tecan Austria GmBH, Grödig, Austria) analyzer.

Ten serum samples of 250 µL were used to determine malondialdehyde (MDA) level according to the method proposed by Kapusta et al. [32]. First, 25 µL of 0.2% 2,6-bis(1,1-dimethylo)-4-methylphenol (BHT, in ethanol) and 1 mL of 5% trichloroacetic acid (aqueous, TCA, Merck, Warsaw, Poland) were added to each sample and vortexed. After centrifugation at 14,000× *g* for 10 min, 750 µL of supernatant was transferred to a glass tube, and 500 µL of 0.6% thiobarbituric acid (aqueous, Merck, Warsaw, Poland) was added, mixed, and incubated for 45 min in a water bath at 90 °C. The supernatants were then stored in cool conditions and centrifuged at 4000× *g* for 5 min. Then, 100 µL of clear supernatant was transferred into a microplate. MDA concentration was determined using Tecan’s NanoQuant Infinite M200 PRO analyzer at the wavelength of 532 nm.

### 2.5. Embryo Bone Measurements

All collected tibias and femurs were cleaned, and their length and diameter in the middle of the shaft were measured using an electronic caliper for linear measurements. The breaking resistance of bones was then determined using a Zwick Testing Machine (Z0.5 Zwicki-Line, Ulm, Germany) with a warhead equipped with a Warner-Bratzel blade with a maximum force of 1 kN. The blade movement speed was 50 mm/min. The six femurs and tibias were weighed on an analytical balance in quartz beakers (weighing approximately 0.1 g), smoked on a hot plate (max. temp. 400 °C), and burned in a muffle furnace with temperature control at approximately 470 °C for 36 h. Subsequently, cooling was performed by adding 2 mL of redistilled water and HCl (38%) to the individual samples. The samples were transferred quantitatively with 40 mL of redistilled water. The mineral content of Ca and P was determined using an ICP-AES Thermo iCAP 6500 DUO (Thermo Fisher Scientific, Waltham, MA, USA) atomic emission spectrometer. Six left femurs were initially grounded in mortar on ice, suspended in 1 mL of PBS, and homogenized. The protein level in sample homogenates was determined using the Bicinchoninic Acid Kit (BCA, Merck, Warsaw, Poland). Then, 25 µL of homogenized supernatants was transferred into a microplate in two replicates, and 200 µL of the reagent mixture was added. The analyzer wavelength was 562 nm. The concentration of OC in homogenates was determined using a chicken-specific enzyme-linked immunosorbent assay (ELISA) kit obtained from Immunogen, Warsaw, Poland (catalog no. MBS268643) and measured using Tecan’s NanoQuant Infinite M200 PRO analyzer at 450 nm according to the protocol. OC and bone alkaline phosphatase (BALP, catalog no. MBS2512530) were also determined in serum.

### 2.6. Broiler Chicken Management

After hatching from eggs previously inoculated with CCN and noninjected ones, 1-day-old chicks from each group were randomly selected for further rearing (90 chicks per group) and not sorted by sex. The next part of the experiment was conducted at the Agricultural Experimental Farm, Wilanów-Obory. The chicks were vaccinated against Marek, Gumboro, and infectious diseases and randomly divided into three replicates of 30 chicks per replicate and placed in rearing boxes with proper dimensions for stocking density. On the 15th day of rearing, vaccination against Gumboro and infectious bronchitis was repeated. The birds were kept on chopped straw litter under standard conditions: The temperature was 32 °C in the first week and lowered by 2 °C weekly to 20 °C in the last week. The humidity was 60%, and 24-h lighting was applied. Microclimate parameters such as humidity and toxic gas content (ammonia, carbon dioxide, and hydrogen sulfide) were recorded at weekly intervals. All parameters were below the standards established by the Regulation of the Minister of Agriculture and Rural Development of 15 February 2010 (Journal of Laws No. 56, item 344) [33]. The birds had free access to water and were fed ad libitum. The applied feed mixtures were starter for chickens at 1–10 days, grower at 11–34 days, and finisher at 35–42 days, formulated in accordance with Ross 308 Nutrition Specifications [34] (Table 1). The BW was recorded at 1, 10, 35, and 42 days of rearing. The feed intake and mortality were registered daily, and final mortality (%) and feed conversion ratio (kg/kg). The mortality, determined by dividing number of dead birds in each group by the initial number of birds in the group and multiplied by 100, FCR = total feed intake/total final BW.

At the end of the experiment, six cocks from each group, having BW within the average BW of the group, were selected, and transferred to a separate slaughterhouse, and euthanasia was performed by decapitation. According to 3rd Local Ethics Committee for Animal Experiments in Warsaw University of Life Sciences, the experiments on chicken embryo and broilers followed the approval of Local Ethics Committee (Approval No. 46/2015). The twelve carcasses were chilled by blowing at 4 °C for 24 h. The cooled carcasses were weighed and then dissected. The contents of breast muscles, leg muscles, heart, liver, and gizzard were determined in relation to BW before slaughter. During decapitation, blood collection was performed.

### 2.7. Meat Quality and Bone Analyses

The pH of breast muscle was measured 24 h after slaughter according to PN-ISO 2917 by using the CP-411 pH meter (Elmetron, Zabrze, Poland) with a combined glass and calomel electrode. The device was previously calibrated in the presence of buffers at pH 4.0 and 7.0. Color parameters were determined for shredded breast muscle using the CR-410 colorimeter (Minolta Co. Ltd., Osaka, Japan) according to the manufacturer’s protocol. Each measurement was performed in two replicates. The values for parameter L* (brightness) were obtained from 0 to 100. Parameters a* and b* are coordinates of trichromaticity. The value +a* corresponds to red, −a* to green, +b* to yellow, and −b* to blue.

The collected and cleaned femurs (right and left) were measured (weight and length), and left bones were prepared for determining breaking resistance using the same method as described above (cf: embryo bone measurements). Subsequently, 3 g of bone (proximal metaphysis and epiphysis area) were obtained by initially grounding in mortar on ice. Ca and P were then evaluated in the fragmented tissue. The concentrations of microminerals (Mg, Mn, Zn, and Cu) were determined using ICP-AES Thermo iCAP 6500 DUO (Thermo Fisher Scientific, Waltham, MA, USA) atomic emission spectrometer was performed. Next, 0.2 g of fragmented femur was lyophilized for 24 h suspended in 1 mL of RIPA buffer (Merck, Warsaw, Poland) for 5 days and homogenized. OC and protein concentration in the homogenates were evaluated (cf: embryo bone measurements).

Right femurs were scanned using a multi-slice 64-row CT scanner (750 Revolution CT, GE Healthcare, Waukesha, WI, USA) following the Gemstone Spectral Imaging (GSI) protocol. The following parameters were used: amperage: ~260 mA; rotation: 0.08/s; HE+: 19.4 mm/rot; slice thickness: 0.6 mm; voltage: GSI-QC (Dual Energy). Images were analyzed with AW VolumeShare7 software (GE Healthcare, Waukesha, WI, USA) in the bone window (W = 2000; L = 350). The auto contour measuring tool was used to fit the measuring window to the bone size in the sagittal, coronal, and axial planes. The current threshold was set at 42 to measure the average bone volume [cm^3^] and the average relative bone density in Hounsfield Units [HU] separately for each bone. The threshold was then adjusted separately for each bone to achieve the average bone volume [cm^3^] for relative bone density of 500 [HU] and the average bone volume [cm^3^] for relative bone density of 1000 [HU].

From the right femur in the area of proximal metaphysis, 0.5-cm-thick fragments were cut, immersed in 10% neutral formalin for 72 h, and decalcified in 15% neutral EDTA buffer (pH = 7.4) (Merck, Warsaw, Poland) for 1 month. The decalcified femurs were dehydrated with graded ethanol (5–100%), defatted in xylene, and embedded in paraffin. Sections of approximately 6 μm thickness were prepared using Microtome Leica RM 2265 (Leica Biosystems, Nussloch, Germany) and used for hematoxylin and eosin (HE) and alizarin red staining. Staining with alizarin red S (Merck, Warsaw, Poland) was performed as follows. First, the rehydrated and defatted sections were stained with 2% alizarin red S solution for 2 min. The sections were rapidly dipped into acetone and acetone xylene (50/50) for 2 s [35]. Microscopic visualization was acquired using a Nikon Eclipse 90i light microscope with a Nikon Digital Sight DS-U1 camera (Nikon Corporation, Tokyo, Japan) and NIS-Elements “D” (Documentation) v.5.02.03 software (Nikon Corporation, Tokyo, Japan). Three visualizations were acquired from each section (18 in total per group) from the repetitive area. The average intensities of alizarin red S staining were quantified using NIS-Elements “D” (Documentation) v.5.02.03 software [36].

### 2.8. Statistical Analysis

The collected data were subjected to statistical analysis using the general linear model of one-factor analysis of variance (ANOVA) with IBM SPSS Statistics v.21.0. The level of significance was set at *p* ≤ 0.05. Individual embryos and birds were treated as experimental units in accordance with the principles of minimum necessary sample size.

## 3. Results

### 3.1. Physicochemical Properties of CCN

The zeta potential of CCN was approximately 20 mV, indicating relative stability. The nanoparticles showed cubic shape with an average size of 15–40 nm. The average size of agglomerate was over 1000 nm (Figure 1).

### 3.2. In Vitro Toxicity and Nineralization Results

Figure 2 shows the results of cytotoxicity test (XTT) of CCN in the concentration range of 5 to 100 μg/mL. The viability of cells was not affected negatively by CCN (no toxic effect). Moreover, the viability of the cells was even higher at higher concentrations of nanoparticles, thus suggesting stimulative effect of CCN on cell viability (high osteoconductive properties). The calcified nodules appeared bright red with higher intensity and density following the increase in the concentration of CCN added to cells, which suggest more effective mineralization processes in these cultures.

### 3.3. In Ovo Results

The results indicated that in IOI of CCN at the concentration of 500 μg/mL did not negatively affect hatchability of the embryos. The hatchability level in both groups was high (over 90%), and mortality caused by infection was negligible. No defects were observed in the developed embryos. The BW of the embryo and breast muscle weight were not affected by CCN. However, the liver weight was affected and was higher in the CCN group (*p* ≤ 0.05) (Figure 3).

The serologic parameters of chicken embryos (Table 2), in most cases, showed no significant differences between the groups, thus indicating normal liver and kidney function. However, MDA concentration was significantly decreased in the CCN group (*p* ≤ 0.05), which may suggest higher peroxidation of lipids. Moreover, the glucose (Glu) concentration was significantly reduced in CCN group.

Figure 4 shows the results of selected measurements performed in the femur and tibia of embryos. No significant differences in bone length, diameter, and breaking resistance were observed between the groups; however, the tibia weight was higher (*p* ≤ 0.05) in the CCN group. The Ca and P concentrations in the femur and tibia were higher in the CCN group (*p* ≤ 0.05), indicating better mineralization in bones of embryos from the IOI group.

### 3.4. Production Results and Meat Quality of Broiler Chickens

The IOI with 500 μg/mL concentration of CCN did not negatively influence the production results and health of the birds from day 1 to day 42 of rearing. The initial BW of birds was similar. There were no differences in the BW on days 1, 10, 35 and 42 (43.3 g vs. 44.1 g, 278.6 g vs. 286.8 g, 2577 g vs. 2558 g, and 3134 g vs. 3147 g for CCN group compared to the control group). The feed conversion rate (FCR) also did not differ significantly between the groups (1.59 vs. 1.54 for the CCN and control groups, respectively). The mortality was at the acceptable level in both groups (Figure 5). The production results were in line with Aviagen recommendations and BW was even higher compared to Ross 308 performance objectives [37].

Table 3 shows the slaughter analysis of male carcasses, where not significant different between the groups. The pH of breast muscles was not significantly different; similar findings were noted for color parameters, except parameter a*. Table 4 shows selected physicochemical properties of breast muscle of broiler chickens.

#### Bone Characteristics of Broiler Chickens

Ca concentration in the femur bone of broiler chickens was significantly higher in the CCN group than in the control group (*p* ≤ 0.05). The relative bone density in Hounsfield scale tended to be higher in the CCN group (*p* = 0.053). The study did not show any significant differences in bone weight, length, and micromineral (Mg, Mn, Zn, and Cu) content (Figure 6).

Figure 7 shows the optical micrographs of the cross sections of the femur (compact and trabecular bone) after alizarin red S and H&E staining. The degree of calcification (intensity of red color) was visibly lower in the bone from the control group as compared to that in the CCN group (average 200.9 vs. 196.9), and the repeatability of measurements was significantly different (*p* ≤ 0.05) (Figure 7). These results show greater amount of alizarin (positive calcified bone) in the CCN group, thus confirming better mineralization in this group.

### 3.5. Molecular Outcomes—OC Concentration in Serum and Femur of Embryo and Broiler Chicken

Figure 8 shows the concentration of OC in the serum and femur of embryos and broiler chickens. In embryos inoculated with CCN, the concentration of OC was lower in the serum but higher in the femur. In broilers, OC levels were not different in the serum; however, CCN birds showed a higher OC concentration in the femur compared to the control group (*p* ≤ 0.05). OC concentration in bones suggests better mineralization.

## 4. Discussion

Nanometric forms of minerals have a high potential to support growth at lower doses compared to conventional organic or inorganic sources of minerals, including Ca [38]. The application of nanotechnology to poultry has been receiving increasing interest because of the continuous search for alternative sources of macro and microminerals that could be more efficiently used in this field. Calcium nanoparticles and their application in poultry production have also become a hot topic of research in recent years [13,16,17,25,39,40,41,42]. Most of the studies have mainly focused on the administration of nanoparticles in water or feed to poultry to improve the health status of birds, productivity, and absorption of minerals. Our study is one of the few research studies investigating the effect IOI of Ca nanoparticles on the skeletal system of birds. The aspect of bird bone quality appears to be important while studying Ca nanoparticles.

The present study applied a high concentration of nanoparticles (500 μg/mL). The in vitro viability assay showed better viability of bone cells at the higher concentration of CCN. We assumed that the applied concentration would not be harmful but more conspicuous, in accordance with our pilot studies. Furthermore, based on our previous study [24], the potential effects of sham control inoculated with PBS were not evaluated in the present work, because of high mortality in the sham control group.

In our previous study [24], the effect of nano-hydroxyapatite (HA-NP) on chicken embryo development, particularly on the skeletal system, was evaluated. The results indicated that HA-NP did not negatively influence embryo development, but influenced molecular responses at the stage of embryogenesis, which were not reflected in bone development of the embryo. In the present study, we decided to use another type of Ca nanoparticles—CCN. The positive effect of IOI of CCN had been already demonstrated by Salary et al. [25] by using the highest concentration of nanoparticles at 200 μg/mL.

The hatchability results of chicken embryos were consistent with those of recent studies by Ahmadzadeh et al. and Matuszewski et al. [24,26] who investigated the use of HA-NP from different origins and CCN [25]. This suggests that CCN, similar to HA-NP, are safe and nonhazardous to the chicken embryo. However, it is worth noting that hatchability was generally lower in the study of Salary et al. [25] even though they used lower concentrations of CCN. The hatchability is influenced by the type of applicated nanoparticles, place and time of inoculation [24], and finally, by the hatching eggs quality [43]. Although the BW of the embryo was not affected, we observed an increase in liver weight. This observation is difficult to explain because of lack of similar studies on this topic; however, CCN may induce differential organ development of chicken embryo [44]. The use of dicalcium phosphate nanoparticles in broiler feed did not affect liver weight [41]. It should be noted that the weight of the bursa of Fabricius and spleen was significantly higher in broiler hatchlings after IOI of CCN in the study of Salary et al. [25]. Chicken embryos are highly sensitive models for testing potential toxic effects of the inoculated substances by monitoring total and differential organ development rates and survivability of the embryos. In the present study, no significant differences were observed in serum biochemical parameters, except MDA, which is a product of lipid peroxidation and Glu. Our results suggest higher lipid peroxidation in the control group; hence, the administration of CCN was found to decrease lipid peroxidation level (which should be considered as a positive effect). It is, however, quite difficult to interpret the relevance of decreased MDA content because of little understanding of the regulation of the antioxidant system in avian embryo. The hydrocolloids of CCN could additionally dilute egg content, resulting in lower lipid peroxidation in embryo. Nevertheless, blood biochemical indicators rarely reflect the actual status of the bird’s health because of their conditioning by many other factors, and in this study, the values for both groups were within the range for poultry [45]. Furthermore, our numerical results agree with those of Ahmadzadeh et al. [26] who demonstrated higher values of AST and ALT in groups after IOI with chemically synthesized hydroxyapatite than in controls. The higher values could be automatically equated with metabolic and liver disfunctions or stress [46]; however, in the present study, we did not observe significant differences between the control and CCN groups. The form and size of the compound supplemented in ovo can affect the organ functions of the embryo. AST and ALT enzymes are normally found in hepatocytes. ALT is a more specific indicator of liver diseases, while AST is commonly found in other organs such as skeletal muscle [47]. In cases of bone metabolism disorders such as phosphatemia, which occur due to decreased ALP levels, bones are poorly mineralized. Thus, serum ALP levels are measured to diagnose bone mineralization and liver functions [48]. In our study, the CCN group showed higher ALP concentration, but the difference was not significant. ALP concentration indicates better mineralization of bones, and results for IOI of CCN obtained by Salary et al. [25] confirmed our findings. Adequate situation was demonstrated according to bacterial synthesized ionic nano-hydroxyapatite applicated to the egg, which influenced more effectively bone mineral content and resulted in higher ALP values of the embryos’ serum [26].

It is well known that bone development dynamics vary in different types of bones. The tibia and femur, for example, show different changes in their structural, mechanical, and compositional properties [49]. The IOI of minerals may affect bone properties. Oliveira et al. [50] showed that the application of organic zinc, copper, and manganese did not affect tibia measurements of 1-day-old hatchlings. Similar results were noted in our previous study—HA-NPs did not affect the width and length of the tibia but affected its weight [24]. The IOI of copper nanoparticles significantly increased femur weight and length in broilers [51]. Ca along with P are the main contributors to bone mineral structure. The mineralization of chicken bones affects their strength, which, in turn, is determined by the mass, volume, microarchitectural organization, and degree of mineralization of bone matrix [52]. The standard indicators of bone mineralization status include various measurements that are partly correlated. These measurements include breaking strength (or resistance to breaking) [53], mineral density [54], crude ash content [53,55], and elemental mineral content, including Ca, P, and microminerals [54]. All these measurements were performed in the present study. The results showed that bone measurements were not affected by IOI of CCN; however, the content of Ca and P in femur and tibia bones was significantly different between the embryo groups. Similar results were also reported by Salary et al. [25], where significant differences in Ca and P concentrations in broiler bones were observed after IOI with 100 and 200 μg/mL of CCN. Moreover, IOI of nutrients increased the Ca and P content of the tibia in embryos on the 19th day of incubation [56]. This is important because almost 99% Ca is derived from skeleton, where it forms hydroxyapatite together with P. In the present study, based on the observed bone quality parameters of broiler chickens, it could be stated that IOI of CCN at 500 μg/mL concentration may cause far-reaching effects—better calcification in broiler chicken’s leg bones. The Ca content in broiler femur was significantly higher in the experimental group than in the control group. However, no differences were observed in the content of other microminerals (Zn, Mg, Mn, and Cu); this finding differed from that of Salary et al. [25] who demonstrated significantly higher Cu values in broiler hatchling bone after IOI of CCN. Increased Ca content may provide higher resistance of bone to breaking, although this aspect was not confirmed in the present study. In broiler chickens, the breaking strength of the femur bone plays an important role in deformities of skeletal system, because this bone is considered to support and perform weight-bearing functions [51]. The bone strength is determined by multiple factors, including bird’s growth rate, age, sex, endocrinal metabolism, bird handling, and feeding [57]. Considering nutritional factors, it is reasonable to pay attention to in ovo feeding using different nutrients through inoculation to improve birds’ bone quality.

The CCN group showed a tendency (*p* = 0.053) of the higher relative femur density compared to the control. This could suggest better mineralization of the femur and actually was reflected in higher Ca concentration in this study. Parameters such as relative bone density and bone volume can differ and depend on factors such as sex, breed, or strain of the chicken [58]. Male chickens are more susceptible to skeletal disorders, especially because they have higher weight gain. Moreover, differences in bone mineral density are influenced by the region where it is measured. In the present study, the measurements of diaphysis, proximal metaphysis, and distal metaphysis were separately performed, but not included. The average density of the entire bone was presented.

The intensity of red color after alizarin red S staining was higher in the femur of the CCN group. Alizarin red S is commonly used in histology and histopathology to stain or locate Ca deposits and Ca-binding proteins and proteoglycans in tissues [59] or cell cultures [60]. Studies on alizarin red S staining usually used rodents as a research model. For example, Fouad-Elhady et al. [36] applied alizarin red S staining to determine the mineralization intensity of the femur bone in a rat model. The degree of calcification was markedly reduced in osteoporotic rats as compared to that in gonad-intact rats, and the maximum amount of alizarin red S (positive calcified bone) was observed in the group treated with HA-NP. The red values were higher in rats than in chicken used in this study.

A few studies have focused on IOI of nutrients in nanometric form and evaluated their effect on after-hatched chicken. In our study, we did not observe any negative effects of IOI of CCN on the final BW, FCR and mortality of chickens after 42 days of rearing. The physicochemical properties of breast muscle were also not affected, despite a change in the color parameter a*, which was probably due to other factors, such as the storage of birds after slaughter. Salary et al. [25] also did not show any significant effect on production results (feed intake, weight gain and FCR) in broilers at 1 to 21 days of age. Other studies on this aspect have been reported, but with IOI of other minerals in nano form. For example, Mroczek-Sosnowska et al. [23] showed the positive influence of Cu nanoparticles and CuSO_4_ on broiler chicken performance. At the end of the rearing period (day 42), the BW was significantly higher in the NanoCu and CuSO_4_ groups than in the control group (2000 g in control vs. 2206 g in NanoCu and 2402 g in CuSO_4_ groups). Both treatment groups had significantly lower FCR and mortality and higher percentage of breast and leg muscles in the carcass than the control group. Several studies have addressed the issue of per os feeding with calcium compound nanoparticles and showed measurable benefits of their use in poultry nutrition. Samanta et al. [39] demonstrated better growth performance in broilers fed calcium phosphate nanoparticles at 50% level. The inclusion of HA-NP in chicken diet led to improved BW gain and feed intake, while Ca and P from HA-NP were better absorbed by birds [42]. Additionally, higher final BW, better FCR, and higher daily weight gains were observed in groups of birds fed diets with nano dicalcium phosphate [41].

Bone modeling is defined as either the formation of bone by osteoblasts or resorption of bone by osteoclasts on a given surface. This contrasts with bone remodeling, in which osteoblast and osteoclast activities occur sequentially in a coupled manner on a given bone surface [61]. The imbalance between these processes might led to the occurrence of osteoporosis—a disease characterized by loss of mineralized structural bones [62,63]. Bone modeling is strictly associated with biochemical bone turnover markers, which play different roles in organisms and are often specific to bone tissue. OC is a major, noncollagenous protein in bones. Recently, this protein has been used as a biomarker of osteoblast activity for evaluating bone remodeling [64]. Few studies have investigated the use of OC as a bone turnover marker in birds. It has been shown that serum OC concentration in laying hens decreases with age. The OC increased in the experimental group (fed with high energy and low protein diet) suggesting greater bone turnover. The quality of bones was not improved, even aggravating the incidence of skeletal damage [65]. Both Ca and P deficiencies reduce hydroxyapatite crystal formation. When bone mineralization decreases, free OC may be available for circulation in the blood [66]. On the other hand, higher OC levels, especially the carboxylated form of OC, may suggest better mineralization process. This dependence was verified in our study, because better mineralization (especially Ca) of bone was observed in embryo and chicken. The CCN group showed higher OC in embryo femur, whereas the control group showed higher concentration of OC in serum suggesting greater bone turnover in control group. In adult birds, serum OC level was similar, but higher in the femur of the CCN group. Because of its nanometric size and easy uptake by cells during embryogenesis, CCN may effectively modulate bone mineralization. Nanoparticles as an external source of Ca allow to reduce the process of hydroxyapatite biosynthesis in osteoblasts and may affect the regulation of bone OC—the protein responsible for hydroxyapatite binding, ultimately resulting in an increase in bone mineralization. This modulation can provide a sustained effect, thereby improving bone quality, even in adult birds. However, further research on this topic is needed.

## 5. Conclusions

It can be concluded that CCN are biocompatible and osteoconductive nanoparticles. The IOI of CCN at the concentration of 500 μg/mL was not harmful to chicken embryos and did not affect hatchability, BW, or muscle weight of the embryos but affected their liver weight. No negative effects were observed for serum biochemical parameters. Moreover, Ca and P content increased in embryo femur and tibia in CCN group. IOI with 500 μg/mL concentration of CCN did not negatively influence the production results and health of the birds from day 1 to day 42 of rearing. The slaughter analysis showed that meat quality (except color parameter a*) of male carcasses was also not affected. Ca concentration in the femur bone of broiler chickens was significantly higher in the CCN group than in the control group, which was a positive aspect. The study did not show any significant differences in bone weight, bone length, and micromineral content in the femur as well as in the scanning results of broiler chickens’ femur. The degree of calcification (intensity of red color) was visibly lower in the bone from the control group than in the CCN group, and the repeatability of measurements was significantly different. The OC concentration in CNN embryos was lower in serum but higher in femur than in the control group. In broilers, CNN group increased OC in femur. The IOI of CCN could modify the molecular responses at the stage of embryogenesis, resulting in better mineralization and could even provide a sustained effect, thereby improving bone quality in adult birds through the calcification of the femur. It can be concluded that IOI with CCN at the concentration of 500 μg/mL did not cause harmful effects and can be used as an alternative method to standard feeding, for improving bone quality of broiler chickens. Further research is needed to determine the possibility of reducing in-feed Ca inclusion in response to IOI of CCN.

## Figures and Tables

**Figure 1 animals-11-00932-f001:**
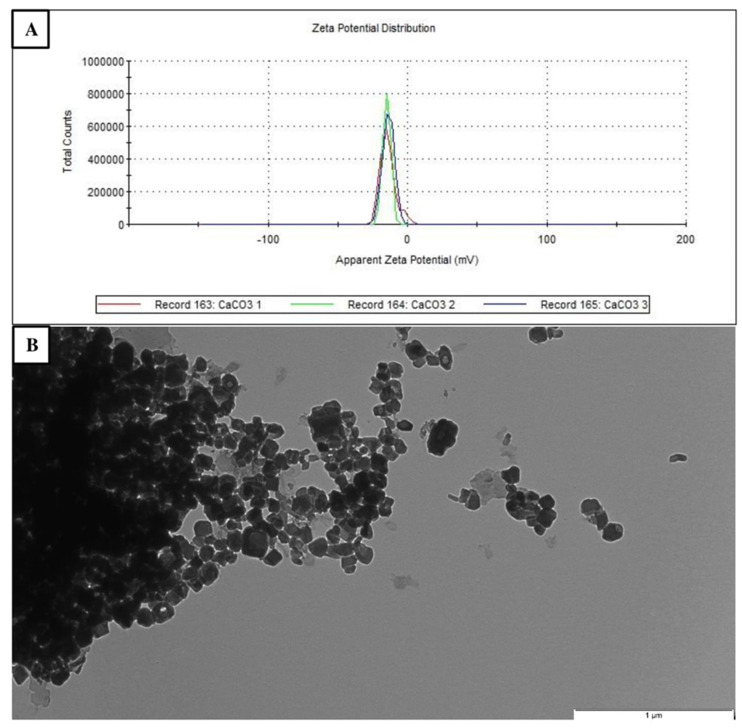
Representative zeta potential of CCN at the concentration of 50 µg/mL (three peaks) (**A**). Transmission electron microscope image of CCN. Scale bar represents 1 µm (**B**).

**Figure 2 animals-11-00932-f002:**
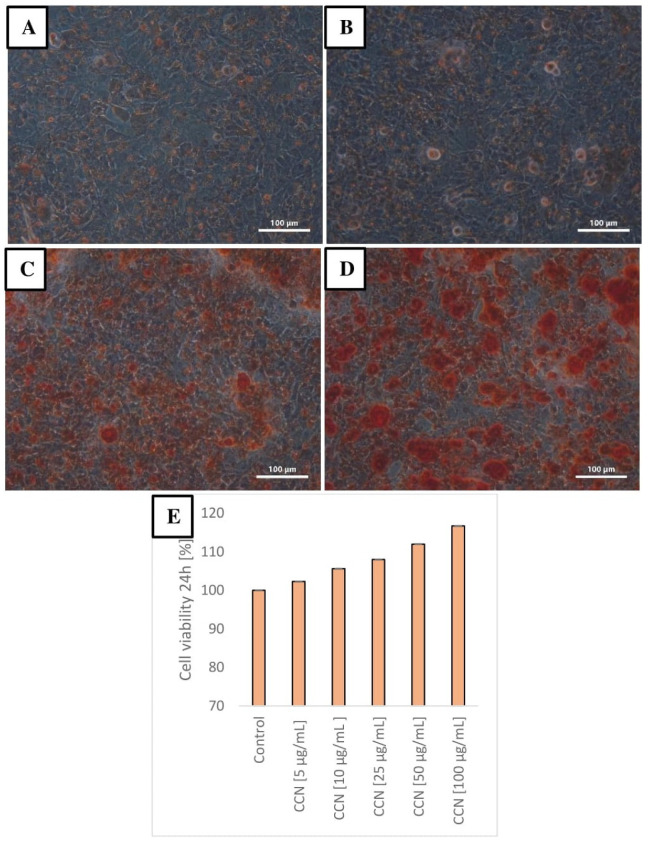
Alizarin red staining for mineralization. The calcified nodules appeared bright red color (original magnification ×100). Cells in control group, without CNN (**A**). Cells incubated with CCN at 5 μg/mL (**B**). Cells incubated with CCN at 25 μg/mL (**C**). Cells incubated with CNN at 100 μg/mL (**D**). Cell viability in groups with increasing CCN concentration determined by the XTT assay after 24 h of incubation (**E**). The error lines represent standard error of mean.

**Figure 3 animals-11-00932-f003:**
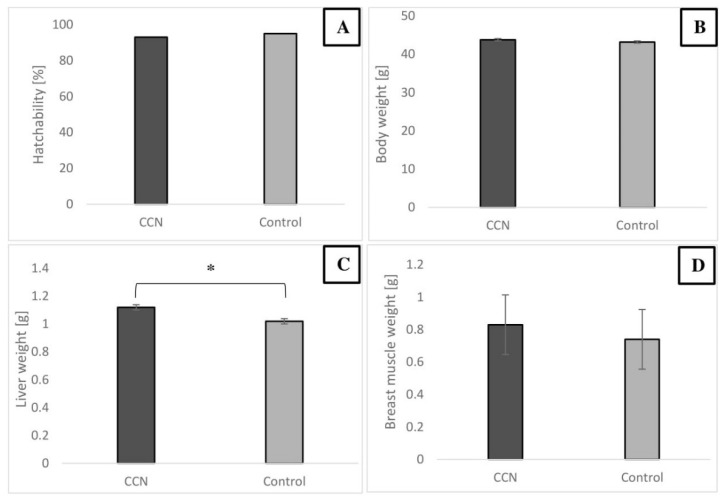
Hatchability (**A**), BW (**B**), liver weight (**C**), breast muscle weight (**D**) of chicken embryos on day 20 after IOI with 500 μg/mL of CCN. * Value on bars differs significantly at *p ≤* 0.05. The error lines represent standard error of mean.

**Figure 4 animals-11-00932-f004:**
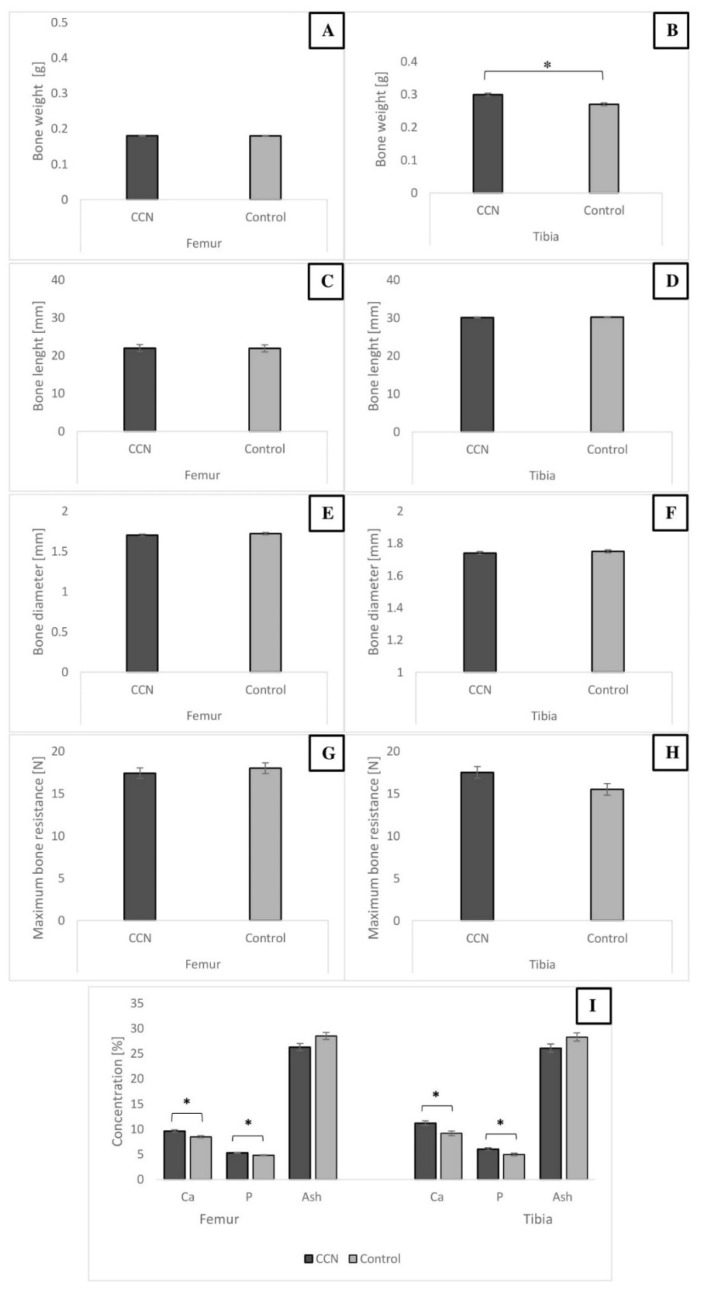
Selected parameters of chicken embryo bones on 20 day after IOI with CNN. Femur and tibia weight (**A**,**B**), femur and tibia length (**C**,**D**), femur and tibia middle diaphysis diameter (**E**,**F**), femur and tibia maximum resistance to breaking (**G**,**H**), femur and tibia Ca, P and ash content (**I**). * Value on bars differs significantly at *p* ≤ 0.05. The error lines represent standard error of mean.

**Figure 5 animals-11-00932-f005:**
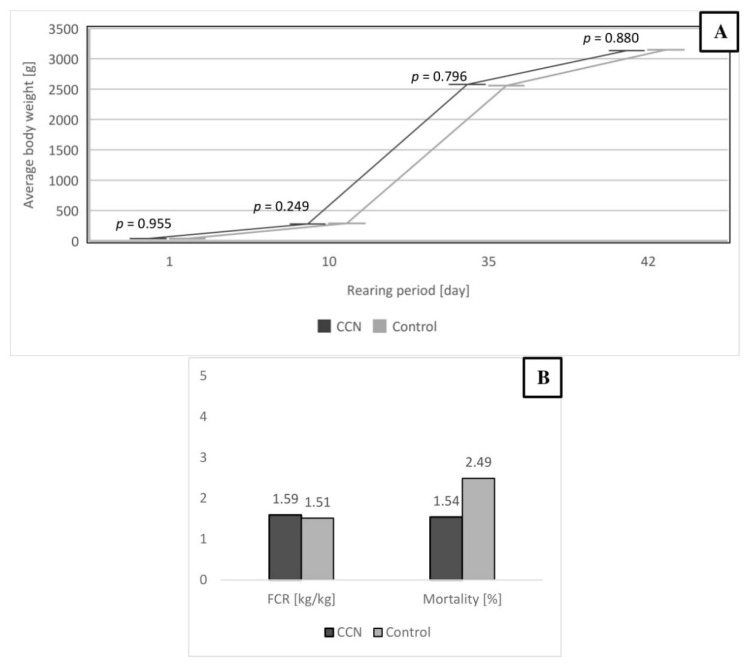
Production results of broiler chickens after 42 d of rearing. Average BW in groups on days 1, 10, 35 and 42. SEM values for average BW on days 1, 10, 35, and 42: 0.239, 3.496, 35.48, and 44.49, respectively (**A**). FCR and mortality of broiler chickens after the rearing period (**B**).

**Figure 6 animals-11-00932-f006:**
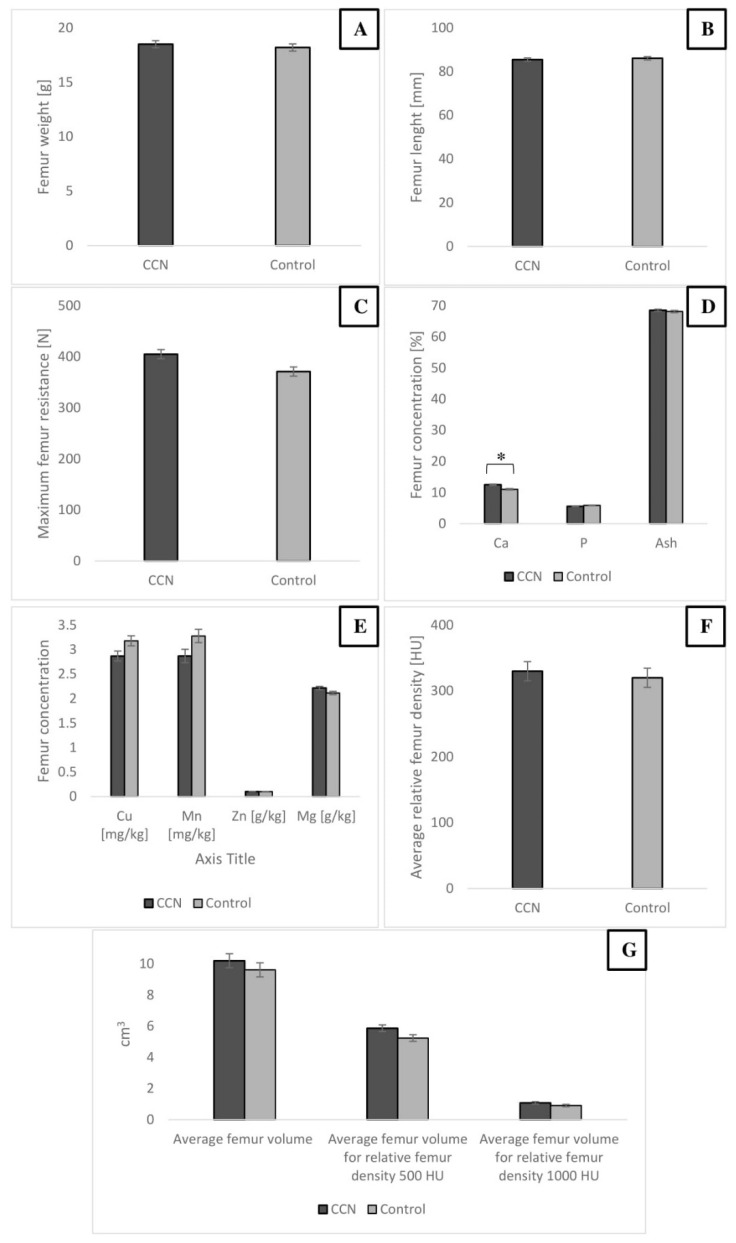
Selected parameters of broiler chicken male femoral bone from different groups. Femur weight (**A**), femur length (**B**), femur maximum resistance to breaking (**C**), femur Ca, P and ash content (**D**), femur micromineral content (**E**), femur average relative mineral density (**F**), average femur volume and average femur volume for 500 and 1000 HU (**G**). * Value on bars differs significantly at *p* ≤ 0.05. The error lines represent standard error of mean.

**Figure 7 animals-11-00932-f007:**
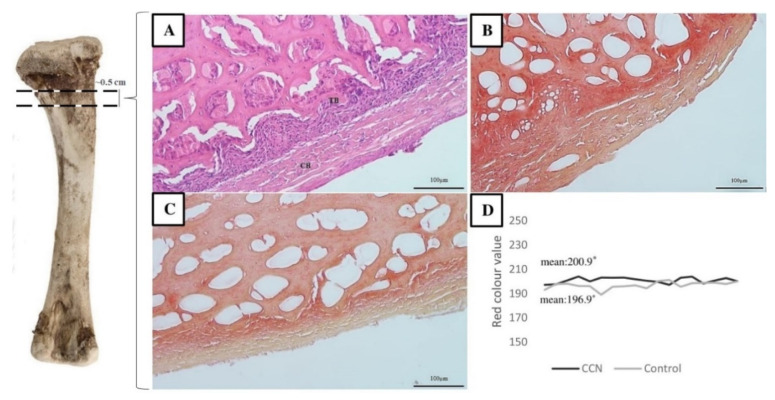
Histological cross sections from proximal metaphysis of broiler chicken male femoral bone. Femur after H&E staining (**A**). Alizarin red staining for mineralization (calcium deposits) for broiler chicken after IOI with CCN (**B**) and without IOI (**C**). Higher red colour intensity value suggests better calcification of the bone (**D**). TB: trabecular bone; CB: compact bone. * The value next to the averages differs significantly at *p* ≤ 0.05.

**Figure 8 animals-11-00932-f008:**
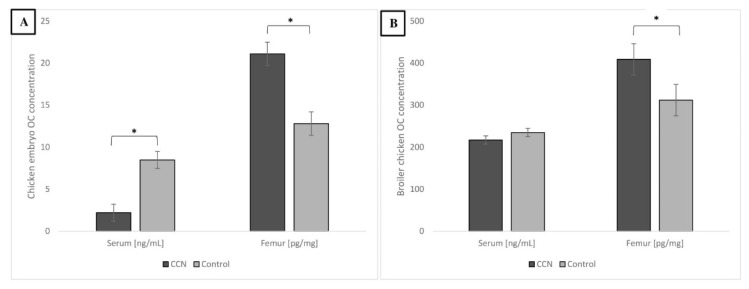
The OC concentration in serum and femoral bone from chicken embryo on 20 day after IOI with CNN (**A**) and from broiler chicken male after 42 d of rearing period (**B**). * Value on bars differs significantly at *p* ≤ 0.05. The error lines represent SEM.

**Table 1 animals-11-00932-t001:** Components and chemical composition of the broiler chicken diets.

Ingredients, g/kg	Starter (Days 1–10)	Grower (Days 11–34)	Finisher (Days 35–42)
Maize	100	114	100
Wheat	530	550	608
Extracted soybean meal	306	274	216
Calcium	11.9	12.0	9.7
Sodium bicarbonate	2.0	1.4	1.6
NaCl	2.4	2.8	2.6
Stimulator	0.1	0.1	0.1
Dicalcium phosphate	11.8	7.8	6.4
Oil	21.0	24.0	44.0
Methionine 84%	4.8	4.2	2.8
Lysine	3.6	3.4	2.8
Threonine	1.4	1.3	1.0
Premix *	5.0	5.0	5.0
**Nutrient Composition, g/kg**			
Analyzed			
Crude protein	219	207	187
Crude fat	47.1	52.3	68.1
Ash	50.7	51.3	51.7
Calculated			
Lysine	12.8	11.8	11.1
Methionine	7.2	6.3	5.4
Calcium	9.5	7.0	7.5
Phosphorus	6.6	5.3	5.1
Metabolisable energy (MJ/kg)	12.28	12.54	12.75

* Rovimix (DSM, Poland): A (retinol acetate) 2,200,000 IU/kg, D3 (E671) 500,000 IU/kg, E (di-alpha-tocopherol acetate 10,000 mg/kg, D (D-pantothenate calcium) 2722 mg/kg, K3 (MNB) 500 mg/kg, B1 (thiamine mononitrate) 400 mg/kg B2 (riboflavin) 1400 mg/kg, B6 (pyridoxine hydrochloride) 800 mg/kg, B12 (cyanocolbalamin) 400 μg/kg, niacin (nicotinic acid) 8000 mg/kg, folic acid 200 mg/kg, biotin 30,000 μg/kg, choline chloride 60,000 mg/kg, copper 1500 mg/kg, zinc (zinc oxide) 11,000 mg/kg, manganese 14,000 mg/kg, iodine (calcium iodate) 120 mg/kg, selenium (sodium selenate) 70 mg/kg, iron (iron sulphate) 9000 mg/kg, citric acid 19 mg/kg, etoxyquin 34.8 mg/kg, propyl gallate 5.4 mg/kg, calcium carbonate 251 g/kg, magnesium 2.2 g/kg.

**Table 2 animals-11-00932-t002:** Serum parameters of chicken embryos.

Parameter	Group			
	CCN	Control	SEM	*p*-Value
AST [U/L]	256	240	31.71	0.716
ALT [U/L]	10.0	6.93	1.774	0.227
ALP [U/L]	948	837	73.43	0.209
BALP [ng/mL]	0.40	0.57	0.053	0.336
Alb [mmol/mL]	15.0	15.3	0.491	0.330
TP [mmol/mL]	24.7	26.0	1.465	0.054
Glu [mmol/mL]	259	276	5.283	0.028
TC [mmol/mL]	139	127	5.564	0.104
TG [mmol/mL]	28.2	32.7	4.066	0.432
LDH [U/L]	1120	1052	98.66	0.413
Cr [mmol/mL]	0.33	0.41	0.022	0.131
MDA [nM/mL]	1.33	1.47	0.022	0.000
GSH [mmol/mL]	3.65	3.30	0.673	0.248
Ca [mmol/mL]	10.5	10.1	0.347	0.248
P [mmol/mL]	7.01	7.58	0.307	0.623

SEM: standard error of mean; CCN: embryos from eggs inoculated with 500 ug/mL hydrocolloid of calcium carbonate nanoparticles; AST: aspartate aminotransferase; ALT: alanine aminotransferase; ALP: alkaline phosphatase; BALP: bone alkaline phosphatase; LDH: lactate dehydrogenase; Glu: glucose; Cr: creatinine; TP: total protein; Alb: albumins; TC: total cholesterol; TG: triglycerides; GSH: glutathione; MDA: malondialdehyde.

**Table 3 animals-11-00932-t003:** Results of male broiler chickens slaughter analysis.

Parameter	Group			
	CCN	Control	SEM	*p*-Value
BW before slaughter [g]	3499	3507	20.21	0.848
Dressing percentage [%]	78.6	78.4	0.207	0.689
Breast muscles [g/100 g BW]	29.2	30.4	0.418	0.136
Leg muscles [g/100 g BW]	19.4	19.3	0.373	0.166
Gizzard [g/100 g BW]	0.82	0.73	0.029	0.102
Liver [g/100 g BW]	2.34	2.12	0.073	0.134
Heart [g/100 g BW]	0.74	0.77	0.035	0.719
Total offal [g/100 g BW]	3.91	3.62	0.080	0.069
Fat [g/100 g BW]	1.64	1.40	0.086	0.181

CCN: embryos from eggs inoculated with 500 ug/mL hydrocolloid of calcium carbonate nanoparticles.

**Table 4 animals-11-00932-t004:** pH and color parameters of breast muscle of male broiler chickens.

Parameter	Group			
	CCN	Control	SEM	*p*-Value
pH	5.78	5.83	0.015	0.128
L*	66.9	64.3	1.151	0.267
a*	14.8	16.4	0.289	0.002
b*	13.5	12.2	0.482	0.198

SEM: standard error of mean; CCN: embryos from eggs inoculated with 500 ug/mL hydrocolloid of calcium carbonate nanoparticles; System L*a*b*, where the L* value designates lightness, ranging from 0 for black to 100 for ideal white, whereas a* and b* are colour coordinates (+a* = redness, –a* = green, +b* = yellow, –b* = blue).

## Data Availability

The data presented in this study are available on reasonable request from the corresponding author.

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
