# Peer review of "Calcium Carbonate Nanoparticles—Toxicity and Effect of In Ovo Inoculation on Chicken Embryo Development, Broiler Performance and Bone Status"

_animals, 2021, doi:10.3390/ani11040932_

Round 1

Reviewer 1 Report

The work is one in a series of studies on the biological effects of in ovo administration of various compounds nanoparticles.

In an extensive experimental setup and using a broad methodological spectrum, the authors assessed the effect of calcium carbonate (CaCO3) nanoparticles (CCN) administered in ovo, on embryo development, bone characteristics and productivity of broiler chickens after 42 days of rearing.

Unfortunately, a slightly careless preparation of the manuscript reduces its value and its content attractiveness. Examples:

  • Introduction. Some vastly general statements poorly explain the rationale behind undertaking the research at hand (“The administration of nanoparticles directly to the egg at the early stage of embryogenesis can lead to a number of molecular and systemic changes. This, in turn, can enable a “better start” for newly hatched chickens and then influence the health and production status of the birds at later stages of life.”)
  • Material and methods.
  • The description is enigmatic and contains numerous understatements that make it difficult to track the course of the procedure and correctness of the methods. The text brevity hinders the results interpretation, or even makes their evaluation impossible.
  • Assessing the potential cytotoxicity and embryo-toxicity of the administered CCN, the authors do not mention anything about the hatchability control and the percentage of embryo death. In the context of the adopted objectives of the study, it seems necessary to include information on the assessment of hatchability and embryo development.
  • There is also no explanation for the lack in the in vivo study of an experimental group for comparison between the effects of CCN and the effects of standard preparations containing calcium carbonate.
  • Point 2.2.1: The description of cell isolation lacks information concerning the type of cells and the method of verification and evaluation of the obtained cell suspension. It makes the the description of the service life test included in point 2.2.2 illegible. It is not known what cells were used in this test. The authors do not mention whether the described procedure was adopted from other authors, or if it was an original idea.
  • In section 2.2.3: "Cell staining": the term "primary osteoblasts" is introduced. Does it apply to the cells isolated according to the procedure outlined in section 2.2.1?
  • Point 2.3 "In ovo experimental design": On what basis do the authors claim that CCN was introduced into fertilized eggs?
  • The authors do not provide the volume of blood taken from the particular embryos. It should be clarified whether the serum comes from a pooled blood, or from an individual blood sample.
  • What is the logical relationship between the statement from verses 157-159 with the description preceding it? (“CCN was administered, incubation was performed and the minimum necessary sample size was estimated with minor modifications, referring to Łukasiewicz et al. and Pineda et al.")
  • References
  • The references should be given in accordance with the editorial requirements.
  • Items: 3, 13, 15, 20, 29, 30, 44, 59 require supplementing and detailed verification.

Author Response

Thank you for the comments. Please find attached our response.

Author Response

(The authors gave the same response as above.)

Reviewer 3 Report

LINES 136 TO 138 – I believe the authors should be certified of this. The embryos are considered sencient after 9 days of incubation because of neural system development. The authors mudt include a letter from the university ethical committee in animal research and teaching, atesting this affirmation.

Institutional Animal Care and Use Committee of the University of Louisville shows a Directive to this activity (http://louisville.edu/research/iacuc/policy-files/UseofChickenAvianEmbryos).

Lines 140 to 144 –In the text: “The eggs were weighed (55.9 ± 1.83 g) on day 1 of incubation and randomly divided into two  groups, with 120 eggs per group. The control group was not injected, and the experimental group was supplemented with 500 μg/mL hydrocolloid of CCN in 0.2 mL volume per egg.  Negative control (injected with PBS) was not included in this study.”

Please, confirm that the inoculation was done in the first day of incubation, because in the text it is not completely clear.

Why do the control group was not inoculated with placebo ? The inoculation effect must be importante in this

Lines 150 to 154 – I understand that these activities must be authorized by the Ethical Committee, as I mentioned in the paragraph above. Some information ara available in an informative of the San Francisco State University (https://research.sfsu.edu/protocol/policy_library/avian_embryo)

Lines 225 to 226 – about the nutritional information: “The applied feed mixtures were starter for chickens at 1-10 days, grower at 11-34 days, and finisher at 35-42 days (Table 1).” The authors must inform which source of nutritional requirements and feed composition They used to formulate the diets.

Lines 242 and 243 – When the authors afirm that: “The experimental procedure followed the approval of 3rd Local Ethics Committee (Approval No. 46/2015).” It seems not logical to do this observation and not approve the procedures to the embryos.

All of them are alive. I understand your University used a national laws to base the attitudes, and I strongly recommend that these texts must be cited in both affirmation about ethical procedures (lines 136 to 138 and 242 to 243).

Line 327 – the term unhatchability seems strange. Explain what do you consider unhatchability?

Lines 333 to 334 – Include the volume of inoculated amount of CCN

Figure 3. Hatchability (A), body weight (B), liver weight (C), breast muscle weight (D) of chicken  embryos on day 20 after IOI with 500 μg/mL of CCN.

Line 400 – Figure 5, please, use the same figure pattern for the figures A and B and show the results (the numbers)

Line 410 – Table 3 – what is Stomach [g/100g BW] ?

Line 414 – These are not physicochemical properties

Line 443 - Figure

Line 465 - Studies conducted to date... Please, it is not easy to understand this phrase.

Author Response

(The authors gave the same response as above.)

Round 2

Reviewer 3 Report

The Figure 5 was corrected and the Ethical guidelines are ok now.